# Leaf Functional Traits in Relation to Species Composition in an Arctic–Alpine Tundra Grassland

**DOI:** 10.3390/plants12051001

**Published:** 2023-02-22

**Authors:** Lena Hunt, Zuzana Lhotáková, Eva Neuwirthová, Karel Klem, Michal Oravec, Lucie Kupková, Lucie Červená, Howard E. Epstein, Petya Campbell, Jana Albrechtová

**Affiliations:** 1Department of Experimental Plant Biology, Faculty of Science, Charles University, Viničná 5, 12844 Prague, Czech Republic; 2Global Change Research Institute, Czech Academy of Sciences, Bělidla 4a, 60300 Brno, Czech Republic; 3Department of Applied Geoinformatics and Cartography, Faculty of Science, Charles University, Albertov 6, 12800 Prague, Czech Republic; 4Department of Environmental Sciences, University of Virginia, Charlottesville, VA 22904, USA; 5Goddard Earth Science Technology and Research (GESTAR) II, University of Maryland Baltimore County, Baltimore, MD 21250, USA; 6Biospheric Sciences Laboratory, Building 33, NASA Goddard Space Flight Center, Greenbelt, MD 20771, USA

**Keywords:** canopy, flavonoids, grasslands, orthophotos, phenolic compounds, remote sensing, secondary metabolism, SLA, species cover analysis, tundra

## Abstract

The relict arctic–alpine tundra provides a natural laboratory to study the potential impacts of climate change and anthropogenic disturbance on tundra vegetation. The *Nardus stricta*-dominated relict tundra grasslands in the Krkonoše Mountains have experienced shifting species dynamics over the past few decades. Changes in species cover of the four competing grasses—*Nardus stricta*, *Calamagrostis villosa*, *Molinia caerulea*, and *Deschampsia cespitosa*—were successfully detected using orthophotos. Leaf functional traits (anatomy/morphology, element accumulation, leaf pigments, and phenolic compound profiles)*,* were examined in combination with in situ chlorophyll fluorescence in order to shed light on their respective spatial expansions and retreats. Our results suggest a diverse phenolic profile in combination with early leaf expansion and pigment accumulation has aided the expansion of *C. villosa*, while microhabitats may drive the expansion and decline of *D. cespitosa* in different areas of the grassland. *N. stricta*—the dominant species—is retreating, while *M. caerulea* did not demonstrate significant changes in territory between 2012 and 2018. We propose that the seasonal dynamics of pigment accumulation and canopy formation are important factors when assessing potential “spreader” species and recommend that phenology be taken into account when monitoring grass species using remote sensing.

## 1. Introduction

The Czech Republic possesses a unique grassland ecosystem in the Krkonoše (Giant) Mountains. Since 1963, the area has been protected as a national park, and it was designated a United Nations Educational, Scientific, and Cultural Organization (UNESCO) biosphere reserve in 1992. Displaying characteristics of both sub-arctic and mountainous regions, the area has been designated Europe’s southernmost relict arctic–alpine tundra [1]. This unique habitat is home to many endemic, glacial relict, and rare species of high conservation value [2,3]. Anthropogenic disturbance has strongly influenced the Krkonoše alpine grasslands, where land was used for grazing livestock from the 9th century until the beginning of the 19th century [4,5]. In the second half of the 20th century, airborne ammonium, nitrate, sulfate, and chloride from air pollution led to widespread acidification of the Krkonoše ecosystems, including the arctic–alpine tundra [6], altering nutrient availability and impacting species composition [7]. Alpine grasslands in general are also experiencing an increasing abundance of shrubs as high-elevation temperatures increase [8,9].

The Krkonoše grasslands have traditionally been dominated by dense tussocks of *Nardus stricta* L.—a short, relatively unpalatable grass known to be present in low-nutrient grazing areas—and provide habitat for endangered invertebrates and ground-nesting birds [2,10]. In the 1970s and 1980s, dolomitic limestone used in paving roads throughout the area resulted in changes in nearby soil pH from 3–3.6 to up to 8 [11]. *Deschampsia cespitosa* (L.) P.Beauv., a species resistant to mechanical damage, began spreading along the edges of hiking paths and roads, where trampling, soil compaction, mineral nutrient content, and increased soil pH created a new microhabitat [12,13]. Additionally, the cessation of traditional grassland management practices (grazing/cutting), combined with pollution-induced soil changes, spruce die-back, increased nitrogen deposition, and warming temperatures, has resulted in the expansion of *Calamagrostis villosa* J.F.Gmel., overtaking *N. stricta* territory [14,15,16]. This tall, defoliation-sensitive grass contributes to losses in species diversity in regions where it becomes dominant [17,18]. Competition among four species (*C. villosa, D. cespitosa, Molinia caerulea* Moench, and *N. stricta*) is of particular interest, as the conservation value of *Nardus* grasslands has become more apparent in recent years [2,19,20].

The growing season in the Krkonoše arctic–alpine tundra grassland is short (snow covers the area 180 days out of the year), and it is characterized by low temperatures in combination with high PAR (photosynthetically active radiation) and UV (ultraviolet) irradiance. High PAR in combination with low temperatures can result in the overproduction of ROS (reactive oxygen species), as the absorbed light energy exceeds what is required to drive photochemistry [21]. At high levels, oxidative stress occurs, with ROS damaging proteins, lipids, and nucleic acids and potentially leading to cell death [22]. Maintaining an equilibrium between the production of ROS and their scavenging by antioxidants is essential to plant survival.

Plants have developed a number of morphological/anatomical, physiological, and biochemical traits as mechanisms to avoid or cope with oxidative stress. These traits represent trade-offs between environmental resource availability and plant growth, survival, and reproduction [23]. Plant functional traits are increasingly used to develop mechanistic models predicting how ecological communities will respond to abiotic and biotic perturbations and how species will affect ecosystem function and services [24]. These can include growth properties that minimize irradiance exposure or increase the boundary layer in order to insulate the plant from the cold and reduce excess transpiration. The contents of chlorophyll and carotenoids indicate the capacity of a plant to engage in photosynthesis and safely dissipate excess photons [25,26]. Phenolic compounds (PheCs)—including anthocyanins, flavonoids, and phenolic acids—function as antioxidants and shield sensitive photosynthetic tissue from UV radiation in order to prevent ROS formation [27,28]. Additionally, PheCs can increase plant tolerance to stressors such as high light, high temperature, drought, heavy metal contamination, herbivores, and pathogens [29,30,31]. The widespread occurrence of PheCs in *Poaceae* makes them valuable chemical markers, as species and even genotypes have been shown to differ in their phenolic profiles [32].

As the climate changes, and microhabitats are created and destroyed, and the adaptive value of certain plant functional traits may also change, leading to changes in species composition [33]. Changes in species cover over the Krkonoše tundra grassland can be monitored remotely using different remote sensing methods, e.g., combinations of airborne and satellite data [34] or via orthophotos [35]. Pigment content, especially chlorophyll, can easily be detected using remote sensing methods, and it can serve as a proxy for plant health [36].

In this study, we compare the morphological, phenological, and biochemical traits of the four competing grass species in Krkonoše. As our work is intended to serve as a pilot study for the monitoring of grasslands using remote sensing (RS), we focused on leaf functional traits easily monitored by RS methods, specifically variations in pigment content among the grass species throughout the growing season and how these variations could influence competition.

The aim of this work was to examine whether the expansion and retreat of grass species could be detected via orthophotos and to determine whether physiological and functional traits correspond to the expansion or decline of these species. We show the varying phenolic profiles of the four main Krkonoše grasses during the vegetative season and show the differences in the accumulation and histochemical localization of PheCs in leaf cross-sections. Furthermore, we present data about the stress status of the different species in homogenous stands. We use chlorophyll fluorescence to evaluate stress and tolerance to tundra conditions, and we present leaf element analysis in order to discuss how some species may be exploiting specific niches. We hypothesize that that interspecific differences in plant functional traits could help to explain the expansion and retreat of grass species in the Krkonoše arctic–alpine tundra.

## 2. Results

### 2.1. Orthophotos Classifications

The main results of the orthophoto classifications from the years 2012 and 2018 are summarized in Table 1. The first area, “U Luční boudy,” meaning, “near the field house Luční bouda,” is a relatively high-traffic area, intersected by tourist walking paths. This area contains *D. cespitosa*, *M. caerulea*, and *N. stricta*, but not *C. villosa*. Wetlands, peat bogs, and water bodies are also present in this area. The overall accuracies for “U Luční boudy” reached ~80% for both years. The second area, “Bílá louka,” meaning, “white meadow,” is more undisturbed, set back from the walking paths and on a slight slope. This area contains *C. villosa, D. cespitosa*, and *N. stricta*, but not *M. caerulea.* It also contains an additional grass species, *Avenella flexuosa* (L.) Drejer. (For more details on the study area, see Section 4.1) The overall accuracies for “Bílá louka” were 78% for 2012 and 85% for 2018 (the relative changes in abundance in each area can be seen in the Figure 1). For both areas, the lowest accuracies for the studied grass species were for *D. cespitosa* due to high variability—sporadic flowering altered the reflectance. Especially in the “Bílá louka” area, the results for *D. cespitosa* were affected by low user accuracy, i.e., the class was overestimated in both years, particularly in 2012, so the decrease in area is lower than 10%. For *C. villosa*, we can confirm that its area increased ~9% between 2012 and 2018, because the class has good accuracy for both producers and users. Producer accuracy is the ratio between the correctly classified pixels and the pixels used for testing of a given class, i.e., real features on the ground. User accuracy describes the probability that a pixel assigned to a class on the map actually represents that class on the ground [37].

*N. stricta* decreased in the first area, “U Luční boudy,” although the decrease could be slightly overestimated, as the user accuracy in 2012 reached only ~78%. *N. stricta* remained stable in the second area, “Bílá louka” (similar accuracies for both years, lower producer accuracies between 75% and 80%). *M. caerulea* achieved good accuracies, apart from the producer accuracy for 2012 (68%), so it could be slightly underestimated in this year; overall we can say that this specie remained stable in the “U Luční boudy” area.

### 2.2. Morphological Characteristics

The main aggressively spreading grass species—*C. villosa* and *M. caerulea* (which is also allegedly spreading [38], although it was not captured by our orthophotos from 2012–2018)—share similar anatomy and morphology. Both are tall grasses possessing flat or convolute leaves (Figure 2a,c, respectively). *C. villosa* has a pronounced midrib and pronounced adaxial segments, especially around larger vascular bundles (Figure 2a). *C. villosa* also notably has a layer of visible anthocyanins below the abaxial epidermis (Figure 2a,e). Details of the bulliform cells (bulbous epidermal cells found in most monocots that allow for leaf movement, i.e., curling of the leaf lamina inward around the midrib) of *C. villosa*, as well as abaxial anthocyanins can be seen in Figure 2f. *M. caerulea* has a smoother adaxial surface and sharper leaf margins than *C. villosa*. *M. caerulea* also has a less prominent midrib (it does not form a sharp point in cross-section as *C. villosa* does). Bulliform cells and small protrusions in the epidermis, which contribute to an increased boundary layer, can be seen in Figure 2h.

*D. cespitosa* appears to be a typical, flat-leaved grass when observed with the naked eye. Under the microscope, however, the structure of *D. cespitosa* is intermediary between the flat leaves of *C. villosa* and *M. caerulea* (Figure 2a,c) and the curled leaves of *N. stricta* (Figure 2d). The leaves of *D. cespitosa* lack an obvious midrib, and they and are divided into triangular segments, each possessing 1–2 vascular bundles with the triangular tip on the adaxial side (Figure 2b). When unstressed, the leaves of *D. cespitosa* open flat or curl slightly outward towards the abaxial side (involute), however the bulliform cells enable the leaves to curl inwards in times of stress, thus lowering the incident radiation and transpirational water loss. Unlike the other species, the bulliform cells of *D. cespitosa* fully connect the adaxial and abaxial leaf surfaces, creating a hinge-like effect that allows for a wide range of motion (Figure 2g). The leaves of *D. cespitosa* are wider than those of *N. stricta*, but they are less than half the width of the other two grasses.

The morphology of *N. stricta* is distinct from the other grasses growing in the Krkonoše tundra grassland. The leaves are small and narrow, macroscopically needle-like, with margins rolled inward to the adaxial (upper) side of the leaf (Figure 1d). The leaves of *N. stricta* develop revolute, without the possibility of fully expanding. As the leaf develops from June to August, the adaxial side of the leaf expands, allowing for slight opening, however the leaves of *N. stricta* remain distinctly curled inward over the adaxial surface throughout their lifetime. Details of the bulliform cells and the abaxial epidermal protrusions are shown in Figure 2i.

Functional leaf traits were measured based on morphological/structural and physiological parameters during the vegetative season in 2020. *C. villosa* has the highest specific leaf area (SLA) of the four species, followed by *M. caerulea*. Both species displayed dynamic changes in SLA over the growing season (Figure 3). The SLA of *C. villosa* decreased at the end of the season (August), while *M. caerulea* reached peak SLA in July and August (Figure 3b). Both *N. stricta* and *D. cespitosa* have low SLA (statistically insignificant from each other), with no discernable variation throughout the growing season (Figure 3).

The percentages of mesophyll tissue (see the green tissue with chloroplasts in Figure 2) and epidermis and vascular bundles (including sclerenchyma as non-photosynthetic structural tissue) were assessed on leaf cross sections of the studied species. *N. stricta* and *D. cespitosa,* with low SLA and rather curled leaves, contained more photosynthetic tissue (mesophyll) on cross-sections than *M. caerulea* and *C. villosa* (Table 2).

### 2.3. Histochemistry

Even without histochemical staining, it was possible to observe the presence of anthocyanins in *C. villosa* on both intact leaves and in cross sections (Figure 2a,e). The visible anthocyanins were localized in the layer of mesophyll cells adjacent to the abaxial epidermis. The distribution of abaxial anthocyanins was not consistent but rather occurred in patches, with clusters of mesophyll cells displaying mainly green chlorophyll in chloroplasts, and other clusters displaying red anthocyanins in vacuoles (see the paradermal view, Figure 2e). Although anthocyanins were detected biochemically in all of the sampled grass leaves (see Section 2.4), only *C. villosa* displayed a concentrated localization of anthocyanins in the leaf, visible to the naked eye as a patchy purple coloration of leaves, and observable as a layer of red-colored mesophyll cells via brightfield microscopy.

Lignin was detected as a pink coloration only in the vascular bundles via phloroglucinol-HCl in all species (Figure 4a,e,i,m); it can also be observed as blue fluorescence in cell walls under UV light (Figure 4c,g,k,o). *C. villosa* showed lignified sclerenchyma supporting minor vascular bundles (Figure 4a); however, lignin only occurred in the vascular bundles (xylem) in the other species (Figure 4e,i,m). The structural tissues present are therefore unlignified sclerenchyma in *D. cespitosa*, *M. caerulea*, *N. stricta*. Condensed tannins, which should appear red when treated with vanillin-HCl, were not found in any species (Figure 4b,f,j,n). The distribution of PheCs is visible as yellow fluorescence when treated with diphenylboric acid 2-amino ethyl ester (DPBA). In *C. villosa* and *M. caerulea*, the PheC distribution was homogenous throughout the mesophyll (Figure 4c,k), and it was more spatially variable in *D. cespitosa* and *N. stricta*. For *D. cespitosa*, there was a stronger adaxial (top) localization of PheC fluorescence; the brightest fluorescence occurred on the tips and sides of the triangular leaf segments that comprise the *D. cespitosa* leaf (Figure 4g,h). By contrast, the abaxial side tended to have stronger fluorescence in *N. stricta*, likely because the adaxial side was curled inwards, so the abaxial (bottom) side receives more exposure to irradiation (Figure 4o,p).

### 2.4. Biochemical Profiles

The total content of anthocyanins, carotenoids, phenolics, and chlorophyll a + b was analyzed using biochemical and spectrophotometrical techniques (see Section 4.3). Significant differences were found for chlorophyll and total anthocyanins, while the total carotenoids and total phenolics did not differ among the species (Figure 5). *N. stricta* accumulated significantly less chlorophyll per leaf dry mass than *D. cespitosa* and *M. caerulea* and fewer total anthocyanins than *D. cespitosa* (Figure 5b). This low chlorophyll content could contribute to the lower palatability of *N. stricta* leaves.

The total contents of anthocyanins, carotenoids, phenolics, and chlorophyll a + b were also evaluated throughout the vegetative season (June–August) in 2020 in order to make observations of the dynamics of accumulation. In *C. villosa,* carotenoids (Figure 6a) and chlorophyll (Figure 6b) showed decreasing concentrations from the start of the season to the end of the season. The total anthocyanins were undynamic (Figure 6c), however, the total phenolics for *C. villosa* significantly increased between June and July, and they remained elevated in August. *C. villosa* was the only species to demonstrate a significant increase in total phenolics; the others maintained relatively fixed concentrations throughout the growing season (Figure 6d). *D. cespitosa* did not show any significant changes in pigment content over the growing season (Figure 6). *M. caerulea* displayed the opposite trend to *C. villosa*, significantly increasing carotenoids (Figure 6a), chlorophyll (Figure 6b), and total anthocyanins (Figure 6c) mid-season. *N. stricta* showed no seasonal dynamics in pigment content (Figure 6).

HPLC-HRMS (high-performance liquid chromatography–high-resolution mass spectroscopy) was used to detect the specific PheCs occurring in the four grasses studied. Fifteen PheCs were consistently found in all samples. Of the main PheCs, five were hydroxycinnamic acids (chlorogenic acid, ferulic acid, caffeic acid, sinapic acid, and 3-coumaric acid), five were hydroxybenzoic acids (3-hydroxybenzoic acid, vanillic acid, syringic acid, protocatechuic acid, and gallic acid), and five were flavonoids (saponarin, luteolin, isovitexin, homoorientin, and apigenin). Due to technical constraints, individual anthocyanins were not identified. *C. villosa* showed high levels of all of the hydroxycinnamic acids tested, except for sinapic acid. *C. villosa* was significantly higher in chlorogenic and caffeic acid levels compared to all other species (Figure 7a,d). For several compounds, *C. villosa* was higher than the other species except *M. caerulea*: ferulic acid, protocatechuic acid, homoorientin, luteolin, and apigenin (Figure 7c,g,h,j,l). *M. caerulea* had significantly higher isovitexin levels compared to the other species (Figure 7b). *M. caerulea* also showed higher sinapic acid levels compared to the other species, except *D. cespitosa* (Figure 7f), but an otherwise low accumulation of phenolic acids. *D. cespitosa* accumulated mainly hydroxybenzoic acids and had higher levels of syringic and gallic acids compared to *N. stricta* and *M. caerulea* (Figure 7k,m) and higher levels of vanillic acid compared to *N. stricta* (Figure 7i). In all cases, *C. villosa* and *D. cespitosa* did not significantly differ in terms of hydroxybenzoic acid accumulation (Figure 7j,k,m). *N. stricta* did not differ from other species in the amount of any tested phenolic compound. While *N. stricta* did show a higher mean saponarin level, due to variability, it did not differ significantly from the other species (Figure 7e). The compounds saponarin, 3-hydroxybenzoic acid, and 3-coumaric acid were accumulated in all species and without significant differences (Figure 7e,n,o).

Principal component analysis was applied to phenolic compounds present in all of the studied species. The results show (Figure 8) that PheCs profiles are species-specific, as the scores clustered according to species. PC1 explained 39% of PheCs variability and was mainly driven by the contents of chlorogenic acid, isovitexin, ferulic, protocatechuic and caffeic acids, luteolin, homoorientin, and apigenin (all positively correlated). The abundance of these compounds is also closely associated with the species *C. villosa*, which separated remarkably from the rest of the species along PC1. PC2 explained an additional 17% of the PheC variability and was driven mainly by the contents of 3-coumaric acid and saponarin (positively correlated) and gallic, syringic, vanillic, and sinapic acids (negatively correlated). *N. stricta* formed a distinct cluster characterized by high contents of p-coumaric acid and saponarin. The 3-hydroxybenzoic acid content contributed to both PC1 and PC2 almost equally. *M. caerulea* scores occupied the position in the center of the coordinate system.

### 2.5. Leaf Element Composition

The leaf element composition averaged for June, July, and August 2020 is presented in Appendix A. The principal component analysis was applied to the element composition of grass leaves in order to reveal whether their nutrition demands contributed to their mutual competition (Figure 9). The first two components explained 74.0% of the variability: PC1 and PC2, 50.6% and 23.4%, respectively. The phosphorus and nitrogen contents and C:N and N:P ratios mainly contributed to PC1. The carbon and calcium contents drove PC2. The potassium and magnesium contents contributed equally to both PC1 and PC2. The species’ separation was not so obvious; however, we still can find some patterns: Mg, C, Ca, N, P, and their ratio (N:P) are the main separators of *D. cespitosa* and *N. stricta. D. cespitosa* separated from the other species mainly due to higher Mg and P contents and low N:P ratio. *N. stricta* scores are located exactly on the opposite side of the PC1–PC2 space, showing low Mg and Ca contents, a high N:P ratio and a rather high C:N ratio. This pattern corresponds to the roles of *N. stricta* and *D. cespitosa* as coexisting but not necessarily competing species. The distribution may be dependent upon microhabitats for *D. cespitosa* (higher Mg and pH demands), as was mentioned in the introduction. The scores for *M. caerulea* and *C. villosa* share the common space, the latter also overlapping with *N. stricta*. This pattern in leaf element composition may indicate that *M. caerulea* and *C. villosa* compete for most nutrients, while for N and P, *C. villosa* competes more with *N. stricta* than *M. caerulea*.

### 2.6. Chlorophyll Fluorescence

The chlorophyll fluorescence parameters Fv/Fm and ΔVip are shown in Figure 10. The parameter Fv/Fm correlates to the maximum quantum yield of photosystem (PS) II photochemistry. It is calculated by dividing variable fluorescence by maximum fluorescence. Fv/Fm has a consistent value of ~0.83 in non-photoinhibited, non-stressed leaves [38,39,40,41]. While all four grass species showed signs of photoinhibition and stress, *N. stricta* had the highest value at 0.73, significantly higher than *D. cespitosa* and *M. caerulea* (Figure 10a). *C. villosa*, *D. cespitosa*, and *M. caerulea* had values of 0.69, 0.67, and 0.67, respectively. The Fv/Fm values were constant over the season for all species except *D. cespitosa*, which had a significant increase in August. The data indicate that *N. stricta* was the least stressed, while *D. cespitosa* and *M. caerulea* were the most stressed.

Another parameter, ΔVip, represents the relative increase in fluorescence between the I and P steps (the thermal phase) in the OJIP curve, correlates with the efficiency of electron flow to photosystem I (PSI) [42], and is higher in samples with greater PSI levels [43]. Elevated ΔVip is associated with environmental stressors, such as salinity [44], drought [45], UV [46], and nutrient deficiencies [43], and it was the highest for *D. cespitosa*. The ΔVip was significantly lower for *C. villosa* and *N. stricta* compared to *D. cespitosa*, while *M. caerulea* did not differ from the other species (Figure 10b), a similar trend to the Fv/Fm. The ΔVip remained constant throughout the season for all of the species except *N. stricta*, which had an insignificant increase in July over June and a significant decrease in August (i.e., stress peaked in July for *N. stricta*).

## 3. Discussion

Species-rich, *N. stricta*-dominated grasslands occur widely throughout Europe and provide essential habitat for rare and endemic species [2]. Long-term pastoral traditions initiated these semi-natural habitats, with grazing animals selectively avoiding the low palatability of *N. stricta* leaves [47]. Recently, changes in land-use [2], land abandonment [48], and increased nutrient availability [49] and soil pH [48], alongside warming temperatures, have threatened the stability of *N. stricta*-dominated grasslands. Data from orthophotos (Figure 1) indicate the retreat of *N. stricta* and the spread of *C. villosa*. They show that *N. stricta* has decreased in total coverage in the highly trafficked U Luční boudy area (although it has not visibly decreased since 2012 in the more remote Bílá louka area). This supports the idea that anthropogenic disturbance plays a partial role in the decline of *N. stricta*. Our data also show an increase in *C. villosa* in Bílá louka (Figure 1). When utilizing remote sensing, results can be influenced by seasonal dynamics. In order to avoid potential phenological changes which might alter classification results, the dates of image acquisition were kept similar (8 July 2012 and 5 July 2018). The final areas of classified species are further influenced by classification accuracies. As Table 1 shows, there are differences in the producer and user accuracies of individual species. The accuracy is influenced by the total area and species’ separability. Since high accuracies were reached for all species (except for *D. cespitosa* in the Bílá louka area), we can reliably confirm the expansion of *C. villosa* and the retreat of *N. stricta*, as was observed by other studies [15,19].

Plant functional traits, such as stem and leaf shape, carbon and nutrient assimilation, chemical defenses, and canopy architecture, all play essential roles in the dynamics of terrestrial vegetation [24,33,49,50,51]. *N. stricta* presents a classically conservative growth form: narrow, erect leaves that reduce exposure to strong irradiance and limit water loss. The bygone selective pressure for efficient nutrient use and leaf durability is compounded by the fact that *N. stricta* is a clonal species with low genetic diversity [52]. *N. stricta’s* strategy of low productivity and long leaf lifespan is seen in its low and non-dynamic SLA (Figure 2). The percentages of structural and photosynthetic tissues in the studied species (Table 2) correspond to differences in SLA; flat and thin leaves, even with higher quantities of structural non-photosynthetic tissues (e.g., for *C. villosa* and *M. caerulea*), may be more effective at light capture in dense continuous canopies [53]. In contrast, low SLA is a trait associated with selective avoidance by some herbivores [54], and the erect leaves reduce high radiative loading and stress [55]. The localization of PheCs on the externally facing leaf side (abaxial in the case of *N. stricta*) is protective against oxidative stress resulting from high PAR and UV irradiation (Figure 4o). PheCs are known to accumulate after exposure to light [55,56,57,58], so the relative lack of PheCs observed on the adaxial side of *N. stricta* leaves may indicate effective structural avoidance of irradiation. By contrast, the flat leaves on *C villosa* and *M. caerulea* have strong PheC fluorescence throughout the mesophyll, indicating steady exposure to high irradiation (Figure 4c,d,k,l).

Tall grasses, particularly *C. villosa* and *M. caerulea*, were previously suppressed by grazing and mowing [15,59,60]. Compared to *N. stricta*, which grows in dense tussocks with erect leaves 25–60 cm high, all three competing grasses have much taller growth forms, capable of overshadowing the shade-intolerant *N. stricta*—*C. villosa* has culms of 15–150 cm, *M. caerulea* of 15–120 cm, and *D. cespitosa* of 20–200 cm [61]. Whether *N. stricta* would have become so dominant without the selective pressure of grazing is uncertain. However, in grasslands where species are mown or cut (a practice which does not discriminate among species), *N. stricta*’s low rate of leaf expansion is uncompetitive [62]. The phenological delay in *M. caerulea* leaf expansion (Figure 3b) may partially explain why this tall species has not spread more (Figure 1a). This is in direct contrast to the early leaf expansion observed for *C. villosa,* which allows it to form a continuous canopy in the absence of mowing/grazing. While many grass species increase tiller density after defoliation, *N. stricta* (in addition to *M. caerulea)* shows a negative relationship between defoliation and tiller density [63]. Mowing slows the spread of *M. caerulea* and *C. villosa*, but it also hinders *N. stricta*. By contrast, *D. cespitosa* was found to expand in grasslands with both sheep grazing [64] and mowing regimes [65].

We were particularly interested in the way biochemical leaf traits may be influencing the dynamics of grass competition in the Krkonoše. Pigments contribute to non-photochemical quenching under high-irradiance conditions (carotenoids [66]) and act as antioxidants (anthocyanins and PheCs [27,30,56]). Chlorophyll, in particular, can be detected remotely and can be used as a proxy for the physiological status of plants [36]. While we did not find any total differences among species in terms of carotenoids or phenolics (Figure 5a,c), chlorophyll content was the highest for *D. cespitosa* and *M. caerulea* and was the lowest for *N. stricta* (Figure 5b). This is clearly reflected for *D. cespitosa* and *N. stricta* in terms of their positions relative to Mg content in the element analysis PCA (Figure 9). This lower concentration is further compounded by *N. stricta*’s higher percentage of photosynthetic tissue per leaf area compared to all other species (Table 2). By contrast, *C. villosa* and *M. caerulea* had the lowest ratio of photosynthetic to non-photosynthetic (structural) tissue (Table 2), i.e., their pigment contents were more concentrated in the leaf mesophyll. In general, “fast carbon acquisition traits,” such as higher SLA and greater chlorophyll content (as in the case of *M. caerulea*), positively correlate with digestibility by ungulate herbivores [67,68].

Regarding anthocyanins, only *D. cespitosa* showed significantly elevated accumulation (Figure 5c). However, the distinct abaxial localization of anthocyanins in *C. villosa* (Figure 2d–f) is a curious feature. Abaxial anthocyanin localization is classically associated with low-light environments and the now-debunked “back-scatter” hypothesis—a theory that abaxial anthocyanins reflected red-photons that had been transmitted through the mesophyll and increased light-capture [69,70]. Adaxial anthocyanins attenuate light, and recent evidence suggests that abaxial anthocyanins function in the same way, preventing photoinhibition, particularly for shade-adapted plants exposed to short, high-intensity light such as transient sun-flecks [30,70,71]. Leaves with adaxial anthocyanins attenuate more light and have reduced photosynthesis at saturating light intensity as compared to those with abaxial anthocyanins [70]. In the case of non-shade plants, abaxial anthocyanins may be a function of leaf orientation: the abaxial side may be more frequently exposed to high irradiance and more sensitive than the adaxial side [72].

Currently, there is no information on how (or if) disperse anthocyanins (i.e., those that cannot be seen in cross sections but can still be detected biochemically, as is the case with *N. stricta*, *D. cespitosa*, and *M. caerulea* [Figure 5 and Figure 6]) affect photoinhibition compared to anthocyanins that localize in a particular cell layer, as with *C. villosa* (Figure 1). Beyond light attenuation and the prevention of photoinhibition, anthocyanin accumulation can also correlate with nutrient stress (P and N deficiencies) [73], particularly when accumulating on the abaxial leaf surface [73,74], and help plants acclimate to cold temperatures [71,75]. Despite having the highest anthocyanin content, *D. cespitosa* simultaneously showed the highest N and P contents among the studied species (Appendix A). However, being a highly plastic species with rather higher nutrient demands [76], we cannot exclude the idea that anthocyanins were accumulated as a response to lower N and P availability.

Throughout the season, pigment concentrations did not significantly vary for *D. cespitosa* and *N. stricta,* while *C. villosa* and *M. caerulea* showed opposite trends. *C. villosa* started the season with a higher SLA (Figure 3b) which decreased over the season alongside carotenoids (Figure 6a) and chlorophyll (Figure 6b), while increasing the concentration of PheCs (Figure 6d). This could potentially enable *C. villosa* to grow quickly and expand early in the season before prioritizing PheC biosynthesis in order to protect its leaves from high irradiation and cold stress for the remainder of the season. *M. caerulea*, by contrast, showed a slight developmental delay compared to the other species. In June, its SLA was still low, increasing significantly by July (Figure 3b) alongside its contents of carotenoids, chlorophyll, and anthocyanins (Figure 6). While individual plants tend to increase their photoprotective pigments after high light exposure, it is interesting that such diverse patterns of pigment accumulation were observed among species over the growing season.

While many PheCs occur across species, specific profiles of PheCs can be indicative of botanical and evolutionary relationships in *Poaceae* [32]. We found distinct clusters of PheCs in the PCA analysis for three of the four species investigated in this study (Figure 8). The profiles of specific PheCs may be more useful than the total phenolics, as the species did not significantly differ in terms of total phenolics (Figure 5d). *C. villosa,* which is expanding its territory in the Bílá louka area (Figure 1), was associated with a diverse range of PheCs: chlorogenic acid, ferulic acid, protocatechuic acid, caffeic acid, luteolin, homoorientin, and apigenin (Figure 8). This wide range of PheCs may protect *C. villosa* from a broader range of environmental stressors than species accumulating only one type of compound. In particular, the dihydroxy B-ring-substituted flavonoids, such as luteolin and apigenin, represent effective antioxidants located within or in the proximity of the centers of ROS generation (in mesophyll cells) in severely stressed plants [57]. Higher diversity of PheCs also ensures the employment of multiple protective mechanisms, which in turn can mean generally higher stability in highly variable environments. *C. villosa* also accumulated high levels of hydroxycinnamic acids (chlorogenic, ferulic, and caffeic acids), which are more efficient antioxidants compared to hydroxybenzoic acids due to their structure [77,78]. By contrast, *D. cespitosa* predominantly accumulated hydroxybenzoic acids—gallic, syringic, and vanillic—at higher levels than the other grasses (Figure 7i,k,m and Figure 8). Grasses accumulating mainly hydroxybenzoic acids may be more responsive to environmental stimuli (ROS also function as signaling molecules) [78], although they may simultaneously be more prone to oxidative stress (this corresponds to our findings with chlorophyll fluorescence, see next paragraph). *N. stricta* was the most associated with a hydroxycinnamic acid (coumaric acid) and a flavonoid (saponarin) (Figure 8), although, due to high variability in *N. stricta* samples, the accumulations of these compounds were not significantly different from the other species (Figure 7e,o). *M. caerulea* did not correlate with a particular group of PheCs according to the PCA (Figure 8), however, it did accumulate significantly more isovitexin than all of the other species (Figure 5b) and significantly more sinapic acid than all of the other species except *D. cespitosa* (Figure 5f). In one comparative study of flavonoids, isovitexin was found to better rescue cells from ROS-induced apoptosis due to its low cytotoxicity, high antioxidative activity, and xanthine oxidase inhibition compared to apigenin, kaempferol, quercetin, myricetin, and genistein [79]. Thus, although not accumulating a wide range of PheCs, *M. caerulea* selectively accumulates an especially protective compound—isovitexin—for the prevention of oxidative stress.

Chlorophyll fluorescence has emerged as a quick, non-invasive technique to evaluate stress in plants [80]. PSII frequently limits photochemistry in response to abiotic stress, especially light. The parameter Fv/Fm corresponds to the maximum efficiency with which light absorbed by PSII is used to reduce Q_A_ (the primary quinine electron acceptor of PSII), and it is frequently decreased when plants are stressed [81]. In general, decreases in Fv/Fm correlate with the stress-induced loss of chlorophyll [80,82]. However, we found the highest Fv/Fm in *N. stricta* (0.73) (Figure 10a), which concurrently had the lowest total chlorophyll content per gram of leaf mass (Figure 5b). Fv/Fm is generally measured to be ~0.83 in non-photoinhibited leaves [39]. Exposure to low temperatures initially induces some photoinhibition, which can also be reversed, as mechanisms such as antioxidant accumulation and xanthophyll cycle activity are increased [83]. The measured values of 0.73 (*N. stricta*), 0.69 (*C. villosa*), and 0.67 (*D. cespitosa* and *M. caerulea*) are in-line with the values observed for cold-hardened plants dealing with stress exposure, and none of these show signs of potentially lethal PSII damage (below 0.4) [83]. The lowest Fv/Fm was measured in *D. cespitosa*, which mainly accumulated PheCs with lower antioxidative capacity (hydroxybenzoic acids) (Figure 8); this supports the findings of other studies that antioxidant capacity is a significant contributor to oxidative damage protection [84]. *D. cespitosa* also showed the highest ΔVip (Figure 10), which is the relative increase between the I and P step of an OJIP curve and is related to electron transfer to PSI end acceptors [42]. A depression of the IP-phase indicates a blockage of the acceptor side of PSI, such as by inactive ferredoxin-NADP+-reductase [85], or lower PSI content [43]. An increase in the PSII:PSI ratio could be created by growing plants hydroponically in a Mg^2+^-deficient solution [43]. It is interesting that Mg content was a main separator of *D. cespitosa* from other species during the PCA of leaves’ elemental compositions (Figure 9). This may indicate higher Mg demands for *D. cespitosa* compared to the other species, resulting in a nutrient-limited inhibition of PSI [86].

*D. cespitosa* was shown to be increasing in abundance in the U Luční boudy area and decreasing in the Bílá louka area, although the accuracies for this measurement were not as high for *D. cespitosa* as for the other species (Table 1). The sampled area neighboring the Luční Bouda chalet was mown and fertilized regularly for at least 250 years until the 1960s, when the area was converted into a strictly protected nature reserve, and almost no direct human management was practiced in the tundra zone. At the same time, tourist pressure intensified, severely disturbing tundra vegetation. This area became known as the “grass garden.” In contrast to the surrounding grassland, the domination of *N. stricta*, *D. cespitosa,* and *Avenella flexuosa* prevailed in the grass garden [87]. The element analysis shows distinct correlation between *D. cespitosa* and Mg content and partly also N and P content (Figure 9). This could be due to the limestone (i.e., calcium magnesium carbonate) used in paving roads in the area creating a distinct microhabitat and the long-term organic fertilization mentioned above, and would negatively influence *N. stricta*, which cannot tolerate calcareous soil [88]. Our results showing different element compositions in *N. stricta* and *D. cespitosa* confirm their contrasting nutrient demands, and, thus, their potential co-existence in localities with heterogeneous nutrient availability [89], although *N. stricta* is better able to tolerate nutrient-poor soils. *D. cespitosa* was also shown to effectively use P in forms of lower biological availability [90], which helps it to coexist with other species in low-nutrient stands. Fertilization significantly increases *D. cespitosa* height, but not that of *N. stricta* [88,91]. In fact, recreating nutrient-poor soil conditions is a recommended practice for restoring *N. stricta*-dominated grasslands [92]. *N. stricta* also prefers drier soil, while *D. cespitosa* thrives under low drainage [93], which may be an additional factor in the decline in *N. stricta* within the moister U Luční boudy area (Figure 1). The value of conservation of species-rich Nardetum communities also includes soil carbon storage, which was higher in Nardetum grasslands with lower P and Ca availability and high herb-species diversity [94].

Leaf phenolic compounds certainly play a role in leaf decomposition and influence nutrient turnover and thus their availability [95]. Phenolic-rich decomposing leaf biomass acts as an allelopathic agent inhibiting competing plant growth and also stimulating microbial activity [96] and N immobilization in microbial biomass [95]. We are aware that the focus on leaf functional traits for disentangling the relationships among grass species is limited without the consideration of belowground processes. However, we preferentially focused on aboveground traits, as this study was conducted as part of a project aimed at interpretating the RS data of different ecosystems. RS methods are crucial for mapping the effects of unprecedented warming on arctic tundra ecosystems [97,98,99]. RS methods are also frequently used for intense vegetation mapping in alpine tundra, since fast changes in vegetation cover occur there as well [99]. Changes in alpine meadow grasslands can be successfully monitored based on multi-source satellite data [100], as we previously showed for the Krkonoše tundra grassland [34,101]. We believe our study will contribute to the RS-based monitoring of ecosystem functioning via the phenological mapping of leaf functional traits in the Krkonoše alpine tundra grasses, as seasonal monitoring of chlorophyll content in the grassland community there has already begun [101]. Further investigations examining the interspecific differences in leaf biochemical traits in relation to leaf chlorophyll fluorescence and canopy solar-induced fluorescence are needed in order to confirm our findings for other species and locations using in-field and remote aerial UAV (unmanned aircraft system) observations.

## 4. Materials and Methods

### 4.1. Study Area

Sampling took place in the relict alpine–arctic tundra grassland of the Giant Mts (50.734 N, 15.696 E, Figure 11) over the years 2020 and 2021. The study area lies 1435 m above sea level. Six plots with a homogeneous canopy were sampled for each of the four species under investigation (24 total). The species’ identities were confirmed by the botanist from the Krkonoše Mountains National Park Administration. Plots were marked with an orientational stake at specified GPS locations. The locations of the plots were detected at the start of each sampling using a Trimble R10 GNSS (global navigation satellite system) device (Westminster, CO, USA). In 2020, chlorophyll fluorescence measurements were taken on-site, and then leaf samples were collected for morphological, biochemical, and element analysis three times per vegetation season in early June, July, and August. The samples for histochemical analyses were taken from fully developed leaves in August 2020. Additionally, the leaf samples for the HPLC-HRMS analysis of individual PheCs were sampled from the identical georeferenced plots in August 2021 in order to extend the insight into PheCs, as the Folin–Ciocalteu assay for soluble phenolic compounds appeared to be too general.

### 4.2. Morphology and Biochemical Analysis

Biomass was collected from two 10 × 10 cm subplots within the main plot. In the laboratory, the collected biomass was divided into fresh biomass and necromass. A subsample of fresh biomass was scanned and then dried in an oven for at least 24 h at 60° and weighed. ImageJ software was used to determine the leaf area from the scans. The dry weight and leaf area of the sample were used to determine SLA (specific leaf area—leaf area related to the unit weight of the dry matter of the leaf) [102]. Many morphological, chemical, and physiological indicators have been proposed for evaluation of the conditions of plants in ecological systems. We selected leaf traits that are connected to photosynthetic capacity (chlorophyll content and chlorophyll fluorescence), protection against PAR and UV (carotenoids, anthocyanins, and PheCs contents) related to the specific conditions of the mountainous site, and SLA related to adaptation to radiative loading and stress. Most of these leaf traits also determine the optical properties of the canopy and are therefore detectable by remote sensing [103,104,105].

The analysis of the prompt fluorescence (PF) of chlorophyll is considered a powerful tool that combines richness of achievable information with operational quickness. This technique is especially suitable for large ecological surveys where it is necessary to screen many samples in a brief span of time. PF analysis allows for evaluation of the photochemical properties and functionalities of photosynthetic organisms utilizing a set of parameters—known as a ‘JIP-test’ and sometimes visualized as an ‘OJIP curve’. The distinct phases of the photochemical processes in terms of energy absorption, trapping, and electron transport can be described by these parameters. In this paper, we reanalyzed large PF datasets obtained from past research carried out under field conditions (forests, plantations, and pasture meadows) and within experimental setups (semi-controlled conditions). Our aim was to explore the relationships (redundancy and independence) among the JIP-test parameters and to select the most suitable ones to capture the variability of plant photosynthetic efficiency and their responses to environmental pressures. Principal component analysis (PCA) was applied to 43,987 measurements. The overall PCA results evidenced that the variability of the PF parameters was mainly explicated by two factors connected, respectively, to the processes of photon capture and primary photochemical events and to the efficiency of the electron transport around Photosystem I. This result suggests that, in ecological studies, the photosynthetic functioning of the member of a population can be effectively described by two parameters representative of these two phases: the maximum quantum yield of the primary photochemistry of a dark-adapted sample (variable to maximal fluorescence intensity Fv/Fm) and the amplitude of the I-P phase (ΔVIP). Fv/Fm and ΔVIP were proven to be independent, and their correlation in various datasets may be either positive or negative in relation to the environmental factors considered. The physiological significance of the correlations between these parameters is discussed [106].

One sample was taken from each of the two subplots in order to determine the anthocyanin and phenolic contents. In order to determine chlorophyll and carotenoids, nine samples were taken per plot. These samples were immediately cooled and stored frozen until further processing. Anthocyanins were extracted in acidified methanol according to the method from reference [107], and they were converted into molar concentration using the Beer–Lambert equation with the universal molar extinction coefficient ε = 30,000 L·mol^−1^·cm^−1^ [108] and related to leaf area. The SLA and molar weight of cyanidin 3-glucoside (one of the abundant anthocyanins in grasses, particularly *M. caerulea* [109]) was used in order to recalculate the quantity of anthocyanins per unit of dry mass into mg. The total content of soluble phenolic compounds was determined from dried and homogenized leaves. Soluble phenolic compounds were extracted in 80% methanol and determined spectrophotometrically using a Folin–Ciocalteu phenol reagent and gallic acid as a standard [110]; final phenol contents were related to dry mass. Chlorophyll and carotenoids were extracted by dimethylformamide [111] and assessed spectrophotometrically, and the contents were calculated according to the method from reference [112] and related to the unit of dry mass.

### 4.3. Histochemistry and Anatomy

All of the samples used for histochemical and anatomical analyses were collected from fully developed leaves in August 2020. Cross sections of grass leaves were made at between 70–80 µm thickness using a hand microtome (Leica RM 2255, Wetzler, Germany). The cross sections were taken from the central part of the blade of fully developed leaves. Phloroglucinol-HCl was used to detect the presence of lignin [113], and vanillin-HCl was used to detect condensed tannins [114] with light microscopy; a 0.1% (*w*/*v*) Naturstoff reagent A solution used to detect flavonoids [115], [116] with fluorescent microscopy utilizing both blue- and UV-light excitation. The presence of anthocyanins was detectable without histochemical staining. All of the samples from all six plots per species were prepared for sections and staining, and the representative samples were presented.

The percentage of mesophyll, epidermis, and vascular bundles including sclerenchyma tissues was assessed using the point-counting method on Naturstoff reagent A-stained fluorescence images. The area of respective tissues on a cross section was related to the area of the whole cross section and was expressed as a percentage.

### 4.4. Specific Phenolic Compounds Identified via HPLC-HRMS

Leaves from six samples of each of the four species (*C. villosa*, *D. cespitosa*, *M. cerulea*, and *N. stricta*) were collected during the 2021 sampling season and were transported back to the lab in plastic bags packed in coolers with ice. The leaves were then lyophilized and stored in vials in silica gel. The samples (0.12–0.27 g of dry mass) were homogenized using a mortar and pestle with liquid nitrogen and were then extracted using a methanol:chloroform:H_2_O solution (v:v:v, 1:2:2). An aliquot of the upper (polar) phase was used to analyze metabolites with an UltiMate 3000 high-performance liquid chromatograph (HPLC) (Thermo Fisher Scientific, US/Dionex RSLC, Dionex, Waltham, MA, USA) coupled with an LTQ Orbitrap XL high-resolution mass spectrometer (HRMS) (Thermo Fisher Scientific, Waltham, MA, USA) equipped with a heated electrospray ionization source. All of the samples were analyzed in the positive and negative polarity of Orbitrap, operated in full-scan mode over a range of *m*/*z* from 50 to 1000 (positive mode) and from 65 to 1000 (negative mode). The PheCs reported in the study were analyzed through non-target analysis, which aims to identify the maximum number of metabolites. PheCs were identified using a standard library and were confirmed by mass, retention time (RT), *m*/*z*, isotope ratios, by checking fragments that were formed during ionization, and also (at higher concentrations) by MS/MS data. During non-target analysis, it was not possible to perform quantification based on the concentration gradient of standards for each of the identified metabolites. Therefore, quantification was performed based on peak area analysis. This analysis allows for relative comparisons between species or variants but does not allow for the evaluation of absolute concentrations, e.g., in µg per g of dry or fresh weight. Over the course of the analyses, an injection of mixed standard (phenolic compounds and phthalates) was then used after analyzing each of the 25 samples. The following PheC groups were identified in all species: hydroxycinnamic acids (chlorogenic acid, ferulic acid, caffeic acid, sinapic acid, and 3-coumaric acid), five hydroxybenzoic acids (3-hydroxybenzoic acid, vanillic acid, syringic acid, protocatechuic acid, and gallic acid), and five flavonoids (saponarin, luteolin, isovitexin, homoorientin, and apigenin). Individual anthocyanins were not identified due to a lack of standards.

### 4.5. Chlorophyll Fluorescence

In 2020, chlorophyll fluorescence was measured using a FluorPen FP 100/S (PSI, Drásov, Czech Republic). First, leaves were dark-acclimated by being wrapped in aluminum foil for 20 min prior to measurement. After the 20 min elapsed, a piece of black velvet cloth was placed over the sample area in order to prevent sunlight from reaching the leaves during measurement. Under the cloth, the aluminum was removed, and the sample was measured using the OJIP function. The OJIP function consists of a saturating pulse of blue light (455 nm, 3000 μmol m^2^/s) being sent to the leaf. The measuring process took place during the interval from 10 μs to 2 s, with a measurement taking place every 10 μs until 600 μs, then every 100 μs until 14 ms, then every 1 ms until 90 ms, and finally every 10 ms until the end of the measurement at 2 s. Two parameters were derived from the measured values and were related to the efficiency of the plant photosynthetic apparatus: Fv/Fm and ΔVip [106]. Five individual leaves were measured at each plot for *C. villosa*, *D. cespitosa*, and *M. caerulea*. The narrowness of the leaves of *N. stricta* necessitated several leaves (4–6) being measured together in order to cover the sensor, and the process was repeated for five tussocks.

### 4.6. Element Analysis

Leaf element analysis was included because element composition and stoichiometry mirror species’ ecological demands and are the subject of competition. Element analyses were conducted on green living biomass dried at 60 °C for 72 h. The analyses were carried out in the certified laboratories of the Research Institute for Soil and Water Conservation (https://www.vumop.cz/en, accessed on 11 November 2022). The samples were redried at 40 °C to constant weight and homogenized. For Ca, Mg, K, and P assessment, the samples were mineralized by HNO_3_ a HClO_4_, and extracts were used for quantification by inductively coupled plasma optical emission spectrometry. The total N content was assessed spectrophotometrically and by Kjeldahl digestion. For total C content, the samples were oxidized with concentrated sulfuric acid and dichromate and finally assessed by titration using thiosulphate.

### 4.7. Statistical Analyses

The differences between species were tested through one-way analysis of variance (ANOVA) at α = 0.05. In the case of normal data distribution, the Tukey–Kramer test was used; otherwise, the Kruskal–Wallis test was used. The analyses were conducted in NCSS 9 software (NCSS 9 Statistical Software (2013). NCSS, LLC., Kaysville, UT, USA, ncss.com/software/ncss). Data for the principal component analysis were centered and scaled using the princomp function from ggfortify package prior to analysis [117]. PCA was accomplished in R (4.1.2) [118].

### 4.8. Analysis of Change in Grass Species Cover Using Remote Sensing

Two sets of RGB aerial data were used—RGB orthophoto from 2012 with a 25 cm pixel size provided by ČÚZK (Czech Office for Surveying, Mapping, and Cadaster) and RGB orthophoto from 2018 with a 12.5 cm pixel size provided by the Krkonoše Mountains National Park Administration. Both orthophotos were acquired at the beginning of July (8 July 2012 and 5 July 2018). Orthophotos from 2018 were resampled to a 25 cm pixel in order to ensure result comparability. Unfortunately, the younger dataset of orthophotos, which would be closer to the sampling dates, was not available. We supervised classification using the Maximum Likelihood algorithm. The training and validation data were collected by botanists in 2020 and were selected from the center of grass patches, so they are not affected by the change. For 2012 and 2018, all polygons were visually checked with the orthophoto and edited where necessary in order to ensure that all of the polygons reliably represented the predetermined categories. One third of the data was used for training and two thirds for validation.

## 5. Conclusions

Our hypothesis—that interspecific differences in plant functional traits could help explain the expansion and retreat of grass species in the Krkonoše arctic–alpine tundra—was upheld by the findings of this research. Despite the similarities between *C. villosa* and *M. caerulea* (i.e., similar leaf morphology, high ratio of structural to photosynthetic tissue, dynamically changing SLA, and high pigment content), only *C. villosa* has been expanding in area. The success of *C. villosa* may be partially explained by its rapid growth and accumulation of pigments early in the season combined its synthesis of a wide array of different phenolic compounds, while *M. caerulea* peaks later and accumulates a narrow field of phenolic compounds. Our findings further suggest that *D. cespitosa* is occupying an available niche (rather than outcompeting neighboring species) due to its accumulation of weaker antioxidants (hydroxybenzoic acids) and greater symptoms of stress (low Fv/Fm and high ΔVip, as well as high anthocyanin content, possibly due to nutrient deficiency). Our analysis of leaf traits reaffirmed *N. stricta’s* suitability in nutrient-poor soils, as it demonstrated low stress levels compared to the other three species. However, the slow growth rate and diminutive form of *N. stricta* present a disadvantage when competing for territory with taller, faster-growing species in the absence of grazing/mowing.

This work also confirmed that changes in grass species’ dynamics can be detected remotely via orthophotos. This study was intended as a pilot study on the use of hyperspectral data acquired by UAVs. We emphasize that the phenology of pigments and morphology throughout the growing season should be taken into account with RS. We hope that hyperspectral data and derived vegetation indices, as well as remote chlorophyll fluorescence and canopy solar-induced fluorescence sensors on UAVs, will be useful in the future to monitor fragile grassland communities.

## Figures and Tables

**Figure 1 plants-12-01001-f001:**
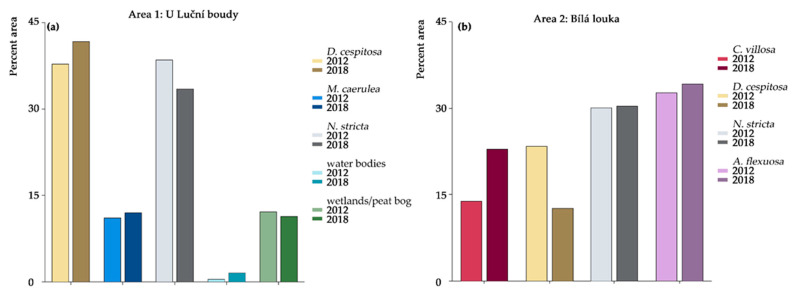
Relative area of the species in both studied areas in 2012 and 2018 based on the maximum likelihood classification of RGB orthophotos with a 25 cm pixel. Area 1: U Luční boudy (**a**); Area 2: Bílá louka (**b**).

**Figure 2 plants-12-01001-f002:**
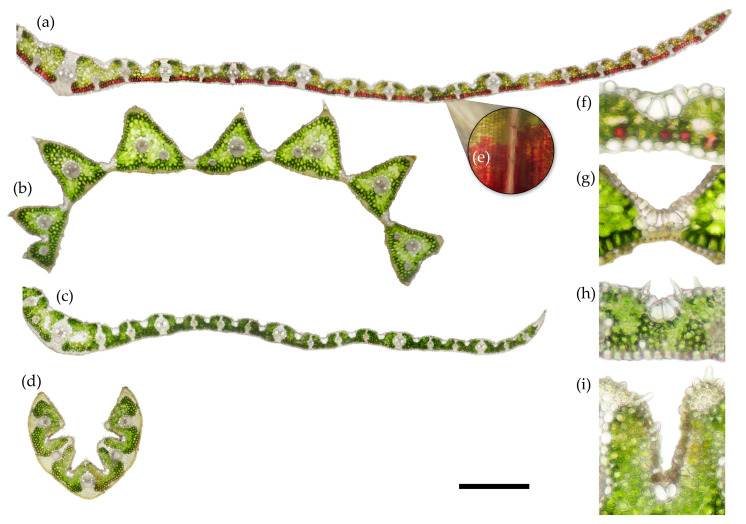
Representative unstained cross-sections of a fully developed leaf from each of the four competing grass species. The section was taken from the central part of the leaf blade in August 2020. Fresh hand sections, 70–80 µm thick, imaged at 10× magnification, bright field; the green color corresponds to mesophyll cells with chlorophyll, while the red color corresponds to vacuolar anthocyanins in mesophyll cells adjacent to abaxial epidermis. *Calamagrostis villosa* (half leaf) (**a**), *Deschampsia cespitosa* (**b**), *Molinia caerulea* (half leaf) (**c**), and *Nardus stricta* (**d**). The scale bar is equal to 0.5 mm. Detail of the anthocyanins in the abaxial mesophyll layer of *Calamagrostis villosa* shown in paradermal view at 10× magnification (**e**). Bulliform cells of each species: *C. villosa* (**f**), *D. cespitosa* (**g**), *M. caerulea* (**h**), and *N. stricta* (**i**).

**Figure 3 plants-12-01001-f003:**
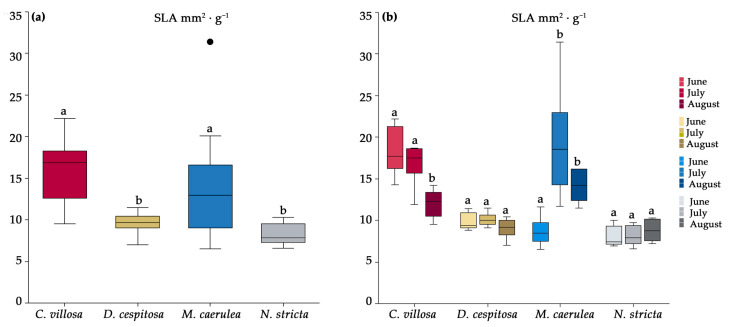
The specific leaf area (SLA) for *Calamagrostis villosa* (red boxes), *Deschampsia cespitosa* (yellow boxes), *Molinia caerulea* (blue boxes), and *Nardus stricta* (grey boxes), measured in 2020 and averaged over the growing season (**a**) and by month (**b**); *n* = 6 per species per month. Medians (central line); 25 and 75 percentiles (boxes); 1.5 interquartile range (error bars); and outliers (dots) are indicated. Different letters denote significance α = 0.05 (one-way ANOVA and Tukey–Kramer post-hoc test).

**Figure 4 plants-12-01001-f004:**
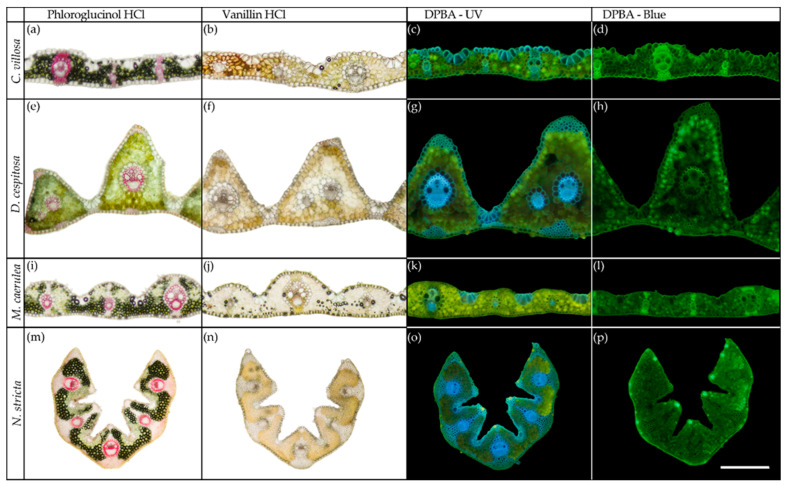
Representative cross-sections of fully developed leaves (August 2020) from the four competing grass species: *Calamagrostis villosa* [first row: (**a**–**d**)], *Deschampsia cespitosa* [second row: (**e**–**h**)], *Molinia caerulea* [third row: (**i**–**l**)], and *Nardus stricta* [fourth row: (**m**–**p**)]. The sections were taken from the central part of the leaf blades in August 2020. Histochemical tests: Phloroglucinol-HCl (first column: **a**,**e**,**i**,**m**) stains lignified cell walls pink, vanillin-HCl (second column: **b**,**f**,**j**,**n**) stains condensed tannins red, Naturstoff reagent A (DPBA) enhances yellow autofluorescence of flavonoids and blue autofluorescence of lignins and cell-wall-bound ferulic acid in UV light (third column: **c**,**g**,**k**,**o**) and (fourth column: **d**,**h**,**l**,**p**) enhances the yellow and green fluorescence of phenolic compounds in blue light excitation. The scale bar is equal to 0.5 mm.

**Figure 5 plants-12-01001-f005:**
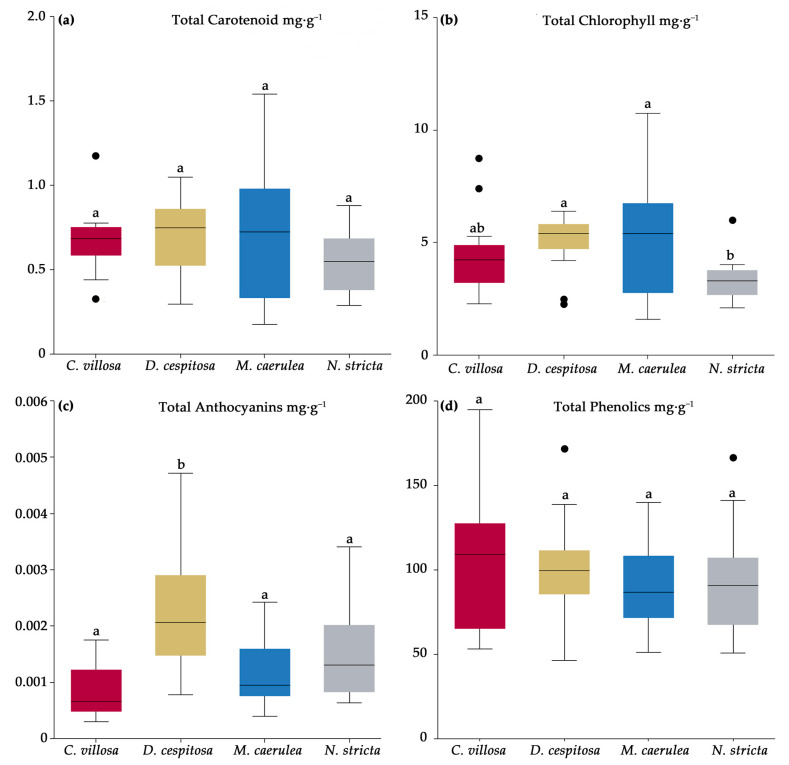
Total pigments of carotenoids (**a**), chlorophylls (**b**), anthocyanins (**c**), and phenolics (**d**) for *Calamagrostis villosa* (red boxes), *Deschampsia cespitosa* (yellow boxes), *Molinia caerulea* (blue boxes), and *Nardus stricta* (grey boxes), measured in 2020 and averaged over the growing season; *n* = 18 per species. Medians (central line); 25 and 75 percentiles (boxes); 1.5 interquartile range (error bars); and outliers (dots) are indicated. Different letters denote significance, α = 0.05 (One-way ANOVA and Tukey–Kramer post-hoc test).

**Figure 6 plants-12-01001-f006:**
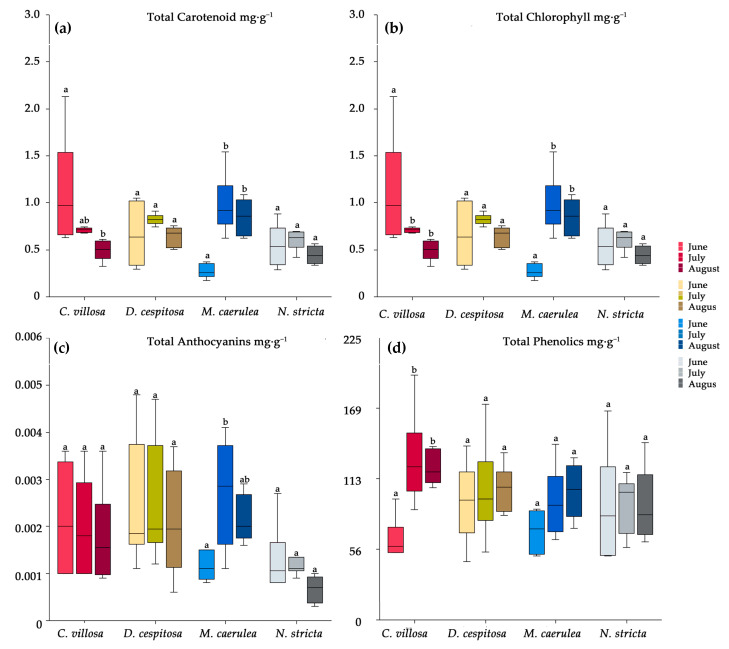
The seasonal dynamics of total carotenoids (**a**), chlorophyll (**b**), anthocyanins (**c**), and phenolics (**d**) for *Calamagrostis villosa* (red boxes), *Deschampsia cespitosa* (yellow boxes), *Molinia caerulea* (blue boxes), and *Nardus stricta* (grey boxes) measured in 2020; *n* = 6 per species per month. Medians (central line); 25 and 75 percentiles (boxes); 1.5 interquartile range (error bars). Different letters denote significance according to the Kruskal–Wallis multiple-comparison Z-value test (α = 0.05) across species.

**Figure 7 plants-12-01001-f007:**
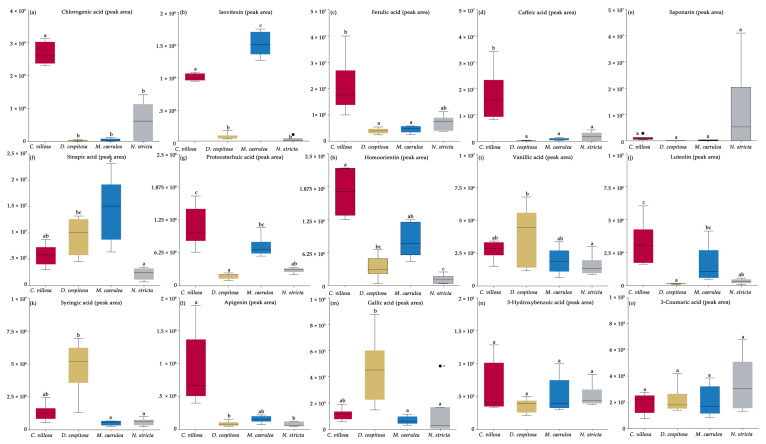
The accumulation of the most abundant phenolics in the four competing grass species from this study: *Calamagrostis villosa* (red boxes), *Deschampsia cespitosa* (yellow boxes), *Molinia caerulea* (blue boxes), and *Nardus stricta* (grey boxes). The PheC levels are expressed as the peak area corresponding to each PheC normalized based on the peak area of respective standard. Chlorogenic acid (**a**), Isovitexin (**b**), Ferulic acid (**c**), Caffeic acid (**d**), Saponarin (**e**), Sinapic acid (**f**), Protocatechuic acid (**g**), Homoorientin (**h**), Vanillic acid (**i**), Luteolin (**j**), Syringic acid (**k**), Apigenin (**l**), Gallic acid (**m**), 3-Hydroxybenzoic acid (**n**), 3-Coumaric acid (**o**). The means are presented (*n* = 6 per species). Different letters of the identical color above boxes indicate statistically significant differences between means (among species) tested by Kruskal–Wallis multiple-comparison Z-value test (α = 0.05) across species.

**Figure 8 plants-12-01001-f008:**
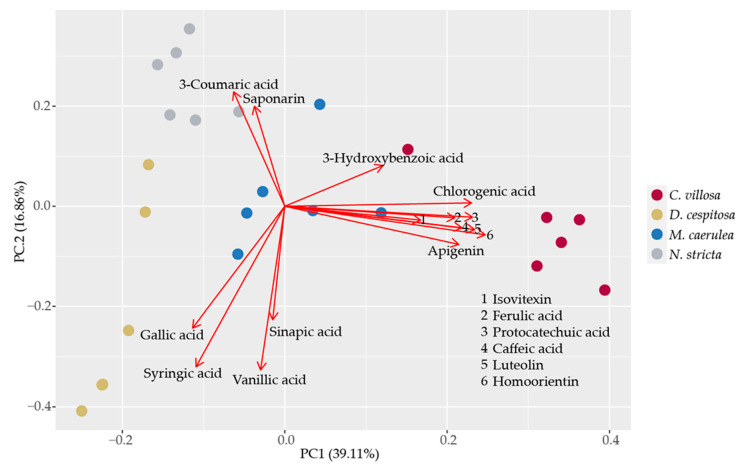
Principal component analysis of the most abundant phenolic compounds in all of the studied species. The points correspond to sample scores and are color-coded by species: *Calamagrostis villosa* (red dots), *Deschampsia cespitosa* (yellow dots), *Molinia caerulea* (blue dots), and *Nardus stricta* (grey dots). Red arrows correspond to PCA loadings for individual PheCs. Data from August 2021 were included in the analysis.

**Figure 9 plants-12-01001-f009:**
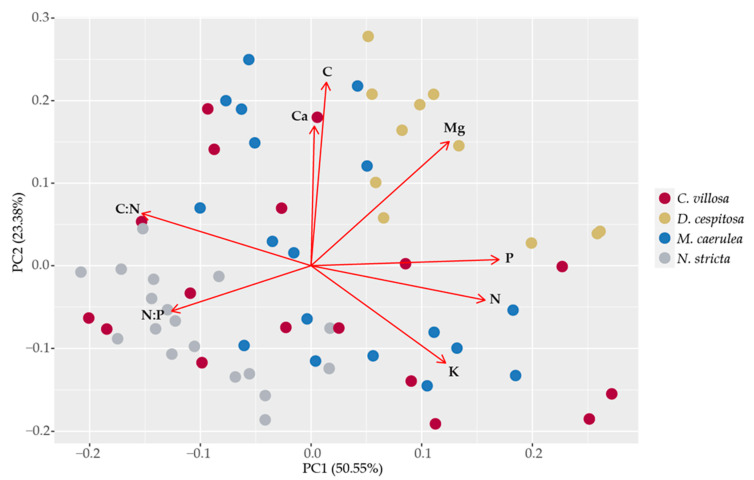
Principal component analysis of the leaf element composition. The points correspond to sample scores and are color-coded by species: *Calamagrostis villosa* (red dots), *Deschampsia cespitosa* (yellow dots), *Molinia caerulea* (blue dots), and *Nardus stricta* (grey dots). Red arrows correspond to PCA loadings for individual PheCs. Data from all vegetative seasons in 2020 were included in the analysis.

**Figure 10 plants-12-01001-f010:**
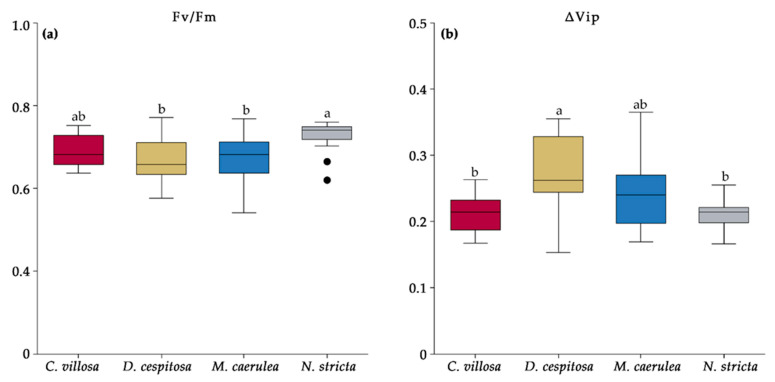
The chlorophyll fluorescence parameters Fv/Fm (**a**) and ΔVip (**b**) measured in the four competing grass species from this study—*Calamagrostis villosa* (red boxes), *Deschampsia cespitosa* (yellow boxes), *Molinia caerulea* (blue boxes), and *Nardus stricta* (grey boxes)—as measured in 2020 and averaged over the growing season; n = 18 (i.e., 6 measurements per species per month—June, July, and August). Medians (central line); 25 and 75 percentiles (boxes); 1.5 interquartile range (error bars); and outliers (dots) are indicated. Different letters denote significance α = 0.05 (one-way ANOVA and Tukey–Kramer post-hoc test).

**Figure 11 plants-12-01001-f011:**
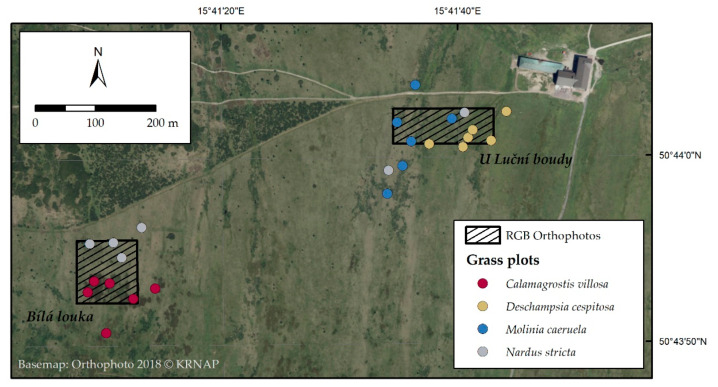
A map of studied area showing the location of sampling plots and two areas where the RGB orthophotos were classified (adapted from Červená et al., 2022 [101]).

**Table 1 plants-12-01001-t001:** Producer and user accuracies for classified species and the area of the species in 2012 and 2018 (PA = Producer accuracy, UA = user accuracy).

Area 1: U Luční Boudy						
class	Accuracies				Area in m^2^	
	2012		2018		2012	2018
	PA	UA	PA	UA		
*D. cespitosa*	79.05	90.84	75.12	85.89	4571.6	5042.9
*M. caerulea*	68.02	90.36	82.78	92.67	1341.9	1445.9
*N. stricta*	92.83	77.61	93.67	89.89	4660.8	4047.4
water bodies	93.44	64.77	70.59	34.53	58.1	183.4
wetlands and peat bogs	73.22	31.02	60.56	26.79	1466.0	1369.6
**Area 2: Bílá Louka**						
class	Accuracies				Area in m^2^	
	2012		2018		2012	2018
	PA	UA	PA	UA		
*C. villosa*	81.07	98.03	92.98	99.25	1628.3	2677.4
*D. cespitosa*	79.74	23.84	55.7	36.26	2751.9	1470.6
*N. stricta*	75.59	97.67	78.94	96.95	3529.0	3546.2
*A. flexuosa*	71.5	35.03	77.48	37.14	3845.2	4006.3

**Table 2 plants-12-01001-t002:** The percentage of mesophyll, epidermis and vascular bundles including sclerenchyma on the leaf cross section. Mean and standard deviation (in brackets); different letters denote significant difference at α = 0.05 according to one-way ANOVA and Tukey–Kramer test. N = 6 for *C. villosa*, *D. cespitosa*, and *N. stricta*; n = 5 for *M. caerulea*.

Species	Mesophyll %	Epidermis %	Vascular Bundles Including Sclerenchyma %
*C. villosa*	46.6 (7.6) b	35.0 (4.9) a	17.9 (4.4) a
*D. cespitosa*	62.0 (4.3) a	30.8 (3.5) a	7.0 (0.6) b
*M. caerulea*	44.0 (3.9) b	33.6 (1.1) a	22.2 (3.3) a
*N. stricta*	60.0 (2.7) a	22.2 (2.2) b	17.5 (1.4) a

## Data Availability

All data are contained within the article or Appendix A.

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
