# Peer review of "Leaf Functional Traits in Relation to Species Composition in an Arctic–Alpine Tundra Grassland"

_plants, 2023, doi:10.3390/plants12051001_

Round 1
Reviewer 1 Report
Plants-2197676 titled "Leaf functional traits of retreating and expanding grass species in a relict arctic-alpine tundra”
My comments on the manuscript are as follow:
1. I have gone through the manuscript thoroughly, the paper is indeed a very novel approach and may be of interest to a wider community and therefore worth reporting. The authors have examined leaf functional traits connected with physiological function and stress detection in four competing species from 2012 and 2018. Further, the authors have used estimates of chlorophyll to compare the relative function and how morphology, element accumulation, leaf pigment and phenolic compounds are associated with spatial expansions and retreats of these four grasses.
2. I don't see any bigger issues, as the overall layout is very clear and conveys a clear message to the readers. The introduction section is enough with a good number of citations, results are appropriate and logically discussed. However, few of the minor comments before the MS is accepted for publication are as under:
a. The title is not clear, and it is very technical, and for specialised/restricted group and it cannot convey a clear message as the aim of the study should. I would suggest modifying it in a way that it has a broader domain and wider audience.
b. All binomials/scientific names must be valid and double checked at: http://www.worldfloraonline.org/, further all scientific names MUST be followed by authority at the place of its first mention.
c. “Figure 2. Representative unstained cross-sections of the four competing grass species. Fresh hand sections…..” needs clarity. Which part is used for section cutting, i.e. leaf/herbaceous stem etc.
d. What is the rational of using only these selected traits, and not others? As there are so many traits, this may be mentioned in an appropriate place in M&M or where it suits.
e. Although, the study is very important and provides a baseline for other studies. But, there are no recommendations made by the study at all? what are the implications of this study this may be added as the concluding remarks in abstract section as well as at the end of the MS. These recommendations MUST be well integrated into futuristic studies/guidelines for following researches; so that other studies/researchers may link their studies to the one here, and may get benefits of this study.
f. I also could not see the ethical approval of the study, who/which body approved this study if the plants contain collection from restricted/reserved areas or the informants and their consent.
g. I could not locate where voucher specimens are submitted and how identification of the specimens was done?
h. Plants valid/accepted names MUST be double checked and compared to the World Flora online database at: http://www.worldfloraonline.org/
i. All abbreviations needs to be described in full at their first place of mention e.g. UNESCO in introduction and others elsewhere in the MS.
j. English language needs to be critically reviewed once more and few sentences are too long, and that obscure the meanings. Such phrases may be shortened and re-structured.
k. Similarity index, formatting and references styles may be double checked and may be consistently followed as per the journal guidelines.
Decision:
While the study is within the scope of the journal, and information may be handy and of wider interest. The MS may be accepted after minor revisions.
Reviewer 2 Report
This is an extensive body of work and the authors have succeeded in presenting the results in an easily digestible format. The manuscript is well written, which also makes it is easy to read. The title is appropriate to the study and the introduction effectively places the work in context. There is however, a problem with the ordering of the sections in the manuscript:
General Comments:
It is unclear why the methods section has been placed after the discussion and before the conclusion. On reading this paper from the beginning it was very confusing to try to understand the results in context of the study setup, without first being offered the methods. As the Results section includes some method information, the reader is drawn to assume that method and results were integrated. I wrote a full page of methodology questions in my report before being surprised to find a methods section buried between Discussion and Conclusions. This section needs to move the usual position following the Introduction.
Having finally found the M&M section there are still some anomalies. The details for the study area and some other methodology detail that appears in the Results section needs to move to this M&M section. Also, there is a lack of clarity regarding when sampling was conducted. The Abstract and start of the results declare that the work was done in 2012 and 2018 yet the data presented in Figure 3 was from 2020. It is further revealed later in the Results section that seasonal pigment production was analysed in 2020, followed by Figure 8 when 2021 data is declared as part of the study. Here again the experimental design is needed to understand why there have been different years when different field and chemical examinations have been made. This suggests that different sets of plants have been used for different tests and as these have been from different growing years. This raises the question of whether any differing conditions in these different years is a factor and if it limits cross reading between data sets. Are the ‘later’ analyses taken from one of the two study areas declared at the start of the Results section, or as a combined sample from both, a range of samples from both or from an entirely different source. There may well be clues within the text, but there is no transparency regarding the experimental setup. The source of the tissue samples shown in the plates is also somewhat obscure about how they link to the other samplings.
Results:
A substantial proportion of the observations are qualitative and not amenable to statistical analyses, but the photographic evidence such as in Figure 4 is excellently presented and very convincing. The use of graphical figures is also very well done giving the reader easy access to an understanding of the observations.
There is not an adequate explanation as to what is meant by Producer’s and User’s accuracies
In Figure 6, it is difficult to understand why some parameters were not significantly different (eg anthocyanins in June versus August for N. stricta.
Discussion:
The discussion is very well written and appropriately provides interpretation and context to the results. From a purist point of view, however, there is no requirement to reference figures in this section. The Results section is well laid out and easy to trawl back into. So (for me) this back referencing is not required and reduces the distinction from the reporting Results and the high-level discussion. This is, of course, an aspect more for the journal’s editor than the authors.
Materials and Methods:
Move to after the Introduction.
Sample numbers range between 5-18, which adds to the confusion around how, what and why certain sampling was imposed or the implication for the analyses, eg expected parameter variance justifying different sample numbers to attain sufficient degrees of freedom to give a stringent test.
Conclusion
It would be more impactful if the authors returned to their hypothesis here and declared as precisely as possible why it was upheld/partly upheld/not upheld, without overly restating the observations. The construction of the current conclusion is more of a summary rather than a conclusion and hence needs some reconfiguring to point to the hypothesis.
